# A discriminator code–based DTD surveillance ensures faithful glycine delivery for protein biosynthesis in bacteria

Santosh Kumar Kuncha[1,2†], Katta Suma[1†], Komal Ishwar Pawar[1], Jotin Gogoi[1], Satya Brata Routh[1], Sambhavi Pottabathini[1], Shobha P Kruparani[1], Rajan Sankaranarayanan[1]*

[1]CSIR–Centre for Cellular and Molecular Biology, Hyderabad, India; [2]Academy of Scientific and Innovative Research, CSIR–CCMB Campus, Hyderabad, India

**Abstract** D-aminoacyl-tRNA deacylase (DTD) acts on achiral glycine, in addition to D-amino acids, attached to tRNA. We have recently shown that this activity enables DTD to clear non-cognate Gly-tRNA$^{Ala}$ with 1000-fold higher efficiency than its activity on Gly-tRNA$^{Gly}$, indicating tRNA-based modulation of DTD (Pawar et al., 2017). Here, we show that tRNA's discriminator base predominantly accounts for this activity difference and is the key to selection by DTD. Accordingly, the uracil discriminator base, serving as a negative determinant, prevents Gly-tRNA$^{Gly}$ misediting by DTD and this protection is augmented by EF-Tu. Intriguingly, eukaryotic DTD has inverted discriminator base specificity and uses only G3•U70 for tRNA$^{Gly/Ala}$ discrimination. Moreover, DTD prevents alanine-to-glycine misincorporation in proteins rather than only recycling mischarged tRNA$^{Ala}$. Overall, the study reveals the unique co-evolution of DTD and discriminator base, and suggests DTD's strong selection pressure on bacterial tRNA$^{Gly}$s to retain a pyrimidine discriminator code.

DOI: https://doi.org/10.7554/eLife.38232.001

*For correspondence:
sankar@ccmb.res.in

†These authors contributed equally to this work

Competing interests: The authors declare that no competing interests exist.

## Introduction

Quality control during translation of the genetic code involves multiple stages and a multitude of proofreading factors (*Guo and Schimmel, 2012*; *Ibba and Soll, 2000*; *Ling et al., 2009*; *Ogle and Ramakrishnan, 2005*). A compromise in editing leads to serious pathologies including neurodegeneration in mouse, and even cell death (*Bacher et al., 2005*; *Bullwinkle et al., 2014*; *Karkhanis et al., 2007*; *Korencic et al., 2004*; *Lee et al., 2006*; *Liu et al., 2014*; *Lu et al., 2014*; *Moghal et al., 2016*; *Nangle et al., 2002*; *Roy et al., 2004*). Among these proofreading factors, D-aminoacyl-tRNA deacylase (DTD) is the one that specifically decouples wrongly acylated D-amino acids from tRNAs (*Calendar and Berg, 1967*; *Soutourina et al., 1999*; *2000*). Our studies have shown that DTD is an RNA-based catalyst that uses an invariant Gly-*cis*Pro motif as a 'chiral selectivity filter' to achieve substrate chiral specificity only through rejection of L-amino acid from the active site, thereby leading to Gly-tRNA$^{Gly}$ misediting (*Ahmad et al., 2013*; *Routh et al., 2016*; *Routh and Sankaranarayanan, 2017*). Recently, we have also shown that DTD's activity on achiral glycine helps in clearing Gly-tRNA$^{Ala}$, a misaminoacylation product of alanyl-tRNA synthetase (AlaRS), thus resolving a long-standing question in translational quality control (*Pawar et al., 2017*).

As DTD acts on multiple tRNAs charged with D-amino acids or glycine, tRNA's role in modulating DTD's activity was previously thought to be inconsequential (*Calendar and Berg, 1967*). However, our recent work has shown that DTD's activity on Gly-tRNA$^{Ala}$ is about 1000-fold higher than on Gly-

tRNA$^{Gly}$, clearly demonstrating the profound effect of tRNA elements on DTD and suggesting an underlying tRNA code for DTD's action. G3•U70, the universal tRNA$^{Ala}$-specific determinant for AlaRS, is also a determinant for DTD. However, it enhances DTD's activity by only about 10-fold (*Pawar et al., 2017*), leaving the 100-fold difference in activity unaccounted for. Here, using exhaustive tRNA sequence analysis and biochemical studies, we identify tRNA's discriminator base (N73) as the major modulator of DTD's activity (unless stated otherwise, DTD refers to bacterial DTD). The invariant uracil (U73) in bacterial Gly-tRNA$^{Gly}$ serves as an anti-determinant and enables the substrate's escape from misediting by DTD, thereby preventing cognate depletion. Using a reporter assay based on green fluorescent protein (GFP), we further demonstrate that DTD is not merely a recycler of Gly-tRNA$^{Ala}$ but avoids glycine misincorporation in proteins. Thus, the study has deciphered the tRNA elements that modulate DTD's activity to ensure faithful delivery of glycine to the ribosomal apparatus during protein biosynthesis. The current work has elucidated why bacterial tRNA$^{Gly}$s—both proteinogenic and non-proteinogenic—have U73 and deviated from harboring purine as N73 during evolution, suggesting for the first time the role of proofreading factors such as DTD in shaping the discriminator base code of tRNAs. The study provides an explanation for a key anomaly in the elegant 'Discriminator Hypothesis' originally proposed by Donald Crothers nearly half a century ago (*Crothers et al., 1972*).

## Results

### U73 is an idiosyncratic feature of bacterial tRNA$^{Gly}$

To identify tRNA elements other than G3•U70 that play a role in modulating DTD's activity, we performed a thorough bioinformatic analysis of all the available 2,671,763 tRNA sequences, which were retrieved from tRNADB-CE (*Abe et al., 2014*). Since DTD is known to act even on D-Tyr-tRNA$^{Tyr}$ digested with T$_1$ RNase (*Calendar and Berg, 1967*), it is not expected to interact beyond tRNA's acceptor stem. Therefore, we focused only on the conservation/variation of acceptor stem bases. The obvious difference is the invariable and unique G3•U70 in tRNA$^{Ala}$, while the rest of the acceptor stem paired bases have no striking partitioning between tRNA$^{Ala}$ and tRNA$^{Gly}$, except the second base pair (*Figure 1b*). However, this is not likely to affect DTD's activity because the second base pair position is not conserved among different tRNAs (e.g., tRNA$^{Phe/Tyr/Trp/Asp}$) on which DTD is known to act with similar efficiency (*Figure 1—figure supplement 1*) (*Calendar and Berg, 1967*; *Soutourina et al., 1999*; *2000*). Surprisingly, bacterial tRNA$^{Gly}$, tRNA$^{His}$ and tRNA$^{Cys}$ have a pyrimidine as N73, while the rest of the tRNAs including tRNA$^{Ala}$ have a purine at that position (*Figure 1a, c*; *Figure 1—figure supplement 2*; *Table 1*). This indicated that N73 could have a role in the differential activity of DTD on Gly-tRNA$^{Gly}$ and Gly-tRNA$^{Ala}$.

### Pyrimidine as N73 acts as an anti-determinant for DTD

To test the role of N73 in modulating DTD's activity, we generated U73A, U73G and U73C mutants of tRNA$^{Gly}$ from *Escherichia coli*. Strikingly, biochemical assays showed that DTD from *E. coli* (EcDTD) deacylates Gly-tRNA$^{Gly}$(U73A) or Gly-tRNA$^{Gly}$(U73G) at 0.1 nM, whereas it shows similar activity on Gly-tRNA$^{Gly}$ at 10 nM (*Figure 2a,c,d*). Thus, EcDTD has nearly 100-fold higher activity on Gly-tRNA$^{Gly}$(U73A) or Gly-tRNA$^{Gly}$(U73G) compared to its activity on the wild-type substrate. In the case of Gly-tRNA$^{Gly}$(U73C), EcDTD's activity at 10 nM is similar to that on Gly-tRNA$^{Gly}$ at the same concentration, thus displaying comparable efficiencies for both substrates (*Figure 2a,b*). This clearly demonstrates that N73 is a key element on tRNA that dictates the efficiency of DTD, and purine as N73 is the preferred base for the enzyme. Our data is further strengthened by the fact that all the reported substrates of DTD, namely D-Tyr-tRNA$^{Tyr}$, D-Phe-tRNA$^{Phe}$, D-Trp-tRNA$^{Trp}$ and D-Asp-tRNA$^{Asp}$, have purine as N73 (*Figure 1c*; *Figure 1—figure supplement 1*) (*Calendar and Berg, 1967*; *Soutourina et al., 1999*; *2000*). This also shows that N73 has a stronger influence on DTD than G3•U70 (which creates only about 10-fold difference) and it can majorly account for the observed 1000-fold difference in DTD's activity on Gly-tRNA$^{Gly}$ and Gly-tRNA$^{Ala}$ (*Pawar et al., 2017*). This further prompted us to test the combined effect of N73 and G3•U70 on DTD's activity.

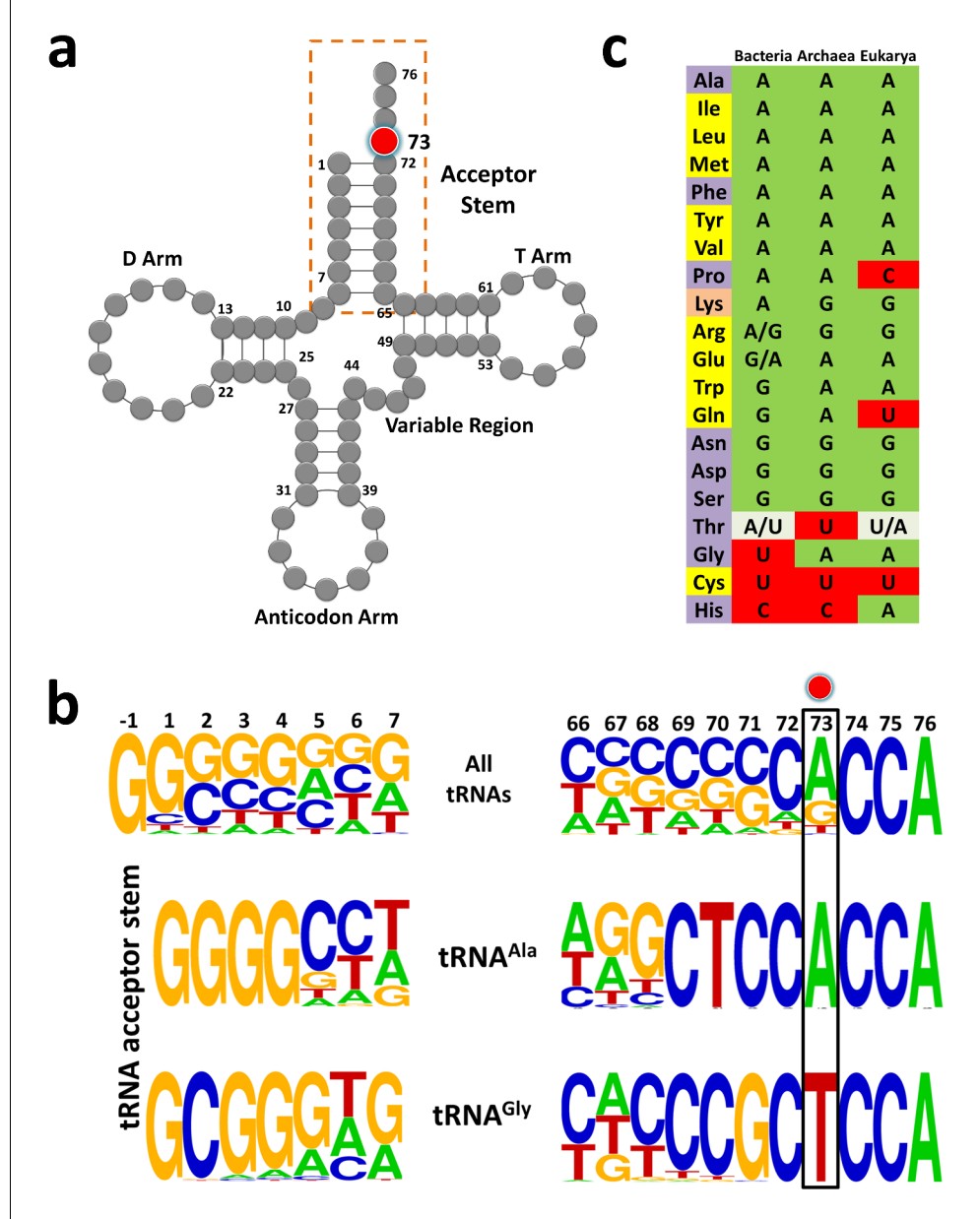

**Figure 1.** tRNA$^{Gly}$ and tRNA$^{Ala}$ show discriminator base dichotomy in Bacteria. (a) Clover leaf model of tRNA with the discriminator base highlighted in red. (b) Frequency distribution of tRNA acceptor stem elements across bacterial tRNAs, comparing and contrasting between tRNA$^{Gly}$ and tRNA$^{Ala}$. Red circle indicates the discriminator base. (c) Distribution of the discriminator base in all tRNAs across the three domains of life. The instances where the discriminator base shows >90% conservation has been represented by the most frequent base. In the case of tRNA$^{Thr}$, U73 and A73 together represent >90% frequency of occurrence; A/U in Bacteria implies A73 is more abundant than U73, whereas U/A in Eukarya denotes U73 is more abundant than A73. Amino acids are color-coded on the basis of the class to which the corresponding synthetases belong: yellow, class I; blue, class II; orange, both class I and II. Discriminator base color-coded as follows: green, purine (A or G); red, pyrimidine (U or C); grey, purine and pyrimidine (A/U or U/A).

DOI: https://doi.org/10.7554/eLife.38232.002

The following figure supplements are available for figure 1:

**Figure supplement 1.** Multiple sequence alignment of a few *E. coli* tRNAs.

DOI: https://doi.org/10.7554/eLife.38232.003

**Figure supplement 2.** Graph showing percentage distribution of the discriminator base in all tRNAs across all bacteria.

*Figure 1 continued on next page*

*Figure 1 continued*

DOI: https://doi.org/10.7554/eLife.38232.004

## N73 and G3•U70 have an additive effect on DTD's activity

To ascertain whether N73 and G3•U70 are mutually independent such that they can have a cumulative effect on DTD's activity, we generated U73A/C70U double mutant of tRNA$^{Gly}$. Biochemical assays showed that EcDTD's activity on Gly-tRNA$^{Gly}$(U73A/C70U) at 0.01 nM is comparable to that on Gly-tRNA$^{Ala}$, and is nearly 1000-fold higher than its activity on wild-type Gly-tRNA$^{Gly}$ (*Figure 2a, e,f*). This unambiguously proves the additive effect of N73 and G3•U70 in modulating bacterial DTD's activity, wherein N73 plays the major role. More importantly, it shows that just two point mutations in the acceptor stem are necessary and sufficient to completely switch the specificity of tRNA$^{Gly}$ to tRNA$^{Ala}$, one which is an anti-determinant in tRNA$^{Gly}$ and the other a positive determinant in tRNA$^{Ala}$. These results also clearly establish that functionally important interactions of DTD are confined only to the acceptor stem of tRNA.

## U73 is the key factor that prevents Gly-tRNA$^{Gly}$ misediting by DTD

Our previous work had shown that elongation factor thermo unstable (EF-Tu) confers protection on Gly-tRNA$^{Gly}$ from DTD's unwarranted activity (*Routh et al., 2016*). tRNA elements on TΨC arm are known to be responsible for EF-Tu binding (*LaRiviere et al., 2001*; *Sanderson and Uhlenbeck, 2007a*; *Schrader et al., 2009*). Hence, N73 mutation in tRNA$^{Gly}$ is not expected to alter the binding affinity of Gly-tRNA$^{Gly}$ to EF-Tu. Moreover, since the effect of U73 as anti-determinant of DTD seen here appeared stronger than the protective effect of EF-Tu observed earlier (*Routh et al., 2016*), we hypothesized that N73 plays the dominant role in preventing misediting of Gly-tRNA$^{Gly}$ by DTD. We

**Table 1.** Table showing the number of tRNAs having a particular discriminator base for all tRNAs across bacteria.

| tRNA$^X$ | A73 | G73 | C73 | U73 | Total |
|---|---|---|---|---|---|
| Ala | 136822 | 9 | 6 | 9 | 136846 |
| Ile | 61982 | 28 | 3 | 3 | 62016 |
| Leu | 259624 | 13 | 9 | 5 | 259651 |
| Lys | 123064 | 122 | 40 | 510 | 123736 |
| Met | 207181 | 12 | 253 | 1868 | 209314 |
| Phe | 68315 | 20 | 2 | 4 | 68341 |
| Pro | 110670 | 9 | 1 | 3 | 110683 |
| Tyr | 80595 | 5 | 6 | 50 | 80656 |
| Val | 187250 | 10 | 7 | 3 | 187270 |
| Arg | 119819 | 103052 | 32 | 10151 | 233054 |
| Glu | 29469 | 54764 | 8 | 46 | 84287 |
| Asn | 20 | 129816 | 6 | 41 | 129883 |
| Asp | 17 | 121556 | 1 | 3 | 121577 |
| Gln | 787 | 102405 | 6 | 2250 | 105448 |
| Ser | 1336 | 192697 | 6 | 3119 | 197158 |
| Trp | 5 | 54387 | 0 | 23 | 54415 |
| Thr | 131288 | 34 | 95 | 28854 | 160271 |
| Cys | 5 | 2 | 8 | 56956 | 56971 |
| Gly | 239 | 11 | 4 | 231855 | 232109 |
| His | 2004 | 4 | 56052 | 17 | 58077 |
| | | | Total bacterial tRNAs: | | 2,671,763 |

DOI: https://doi.org/10.7554/eLife.38232.005

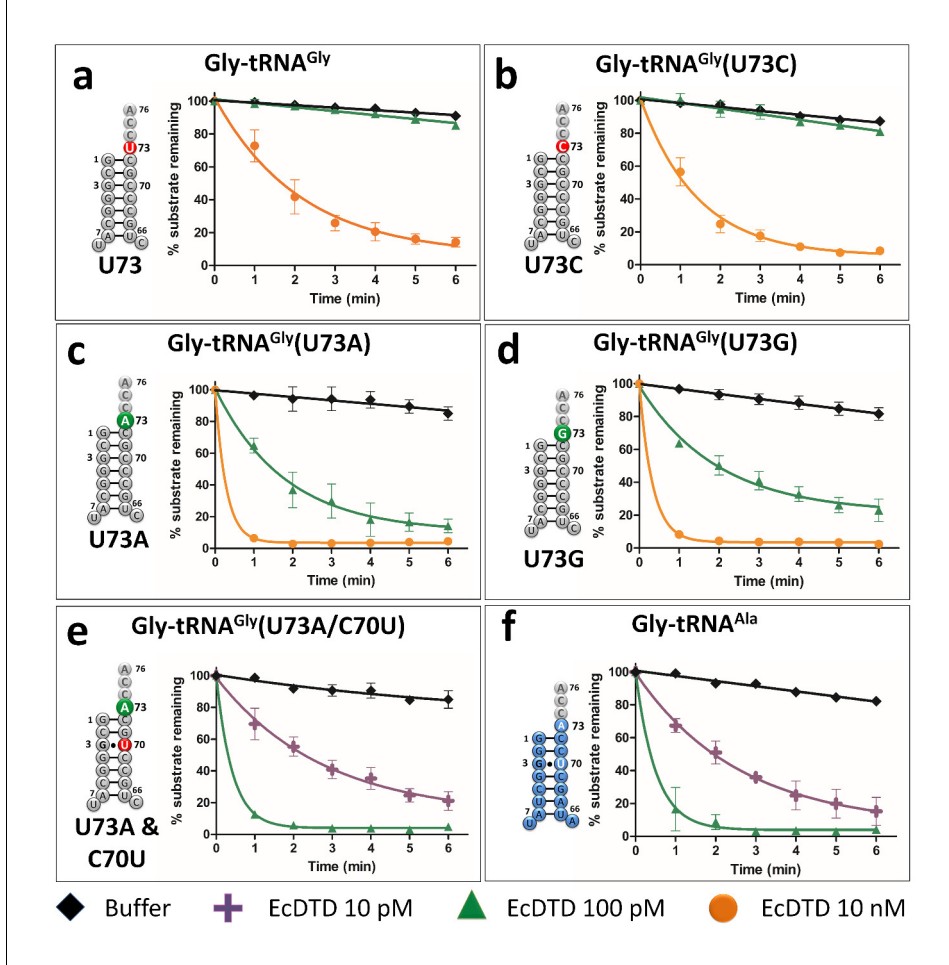

**Figure 2.** Discriminator base modulates DTD's activity. (**a–e**) Deacylation of Gly-tRNA[Gly] and its mutants by various concentrations of EcDTD. (**f**) Deacylation of Gly-tRNA[Ala] by various concentrations of EcDTD. Lines indicate exponential decay fits and error bars represent one standard deviation from the mean of at least three independent readings.

DOI: https://doi.org/10.7554/eLife.38232.006

The following source data is available for figure 2:

**Source data 1.** Biochemical data for EcDTD deacylations with Gly-tRNA[Gly/Ala] (wild type and mutants).

DOI: https://doi.org/10.7554/eLife.38232.007

probed this by performing competition experiments between EF-Tu and EcDTD. In the absence of EF-Tu, EcDTD acts on Gly-tRNA[Gly] at 10 nM, while the presence of EF-Tu protects Gly-tRNA[Gly] from 10 nM EcDTD. The substrate is, however, completely deacylated by 100 nM EcDTD even in the presence of EF-Tu. Thus, the protection offered by EF-Tu to Gly-tRNA[Gly] against EcDTD is less than 10-fold (*Figure 3a*), which is in agreement with our previous study (*Routh et al., 2016*).

In the case of Gly-tRNA[Gly](U73A), EcDTD's activity without EF-Tu is at 0.1 nM, while in the presence of EF-Tu, similar activity is observed at 1 nM (*Figure 3b*). Hence, irrespective of the identity of N73, EF-Tu offers only about 10-fold protection. Furthermore, the 100-fold difference in the activity of EcDTD on Gly-tRNA[Gly](U73A) and wild-type Gly-tRNA[Gly] is maintained even in the presence of EF-Tu. These findings clearly demonstrate that EF-Tu has no preference for N73, and it is DTD's differential activity that predominantly determines the fate of the substrate. Thus, EF-Tu augments the protection of Gly-tRNA[Gly] primarily/majorly conferred by U73, that is the enhancement in protection of Gly-tRNA[Gly] by EF-Tu is consequential only if the substrate harbors U73. Nevertheless, cellular DTD levels must be tightly regulated because DTD overexpression has been shown to cause toxicity

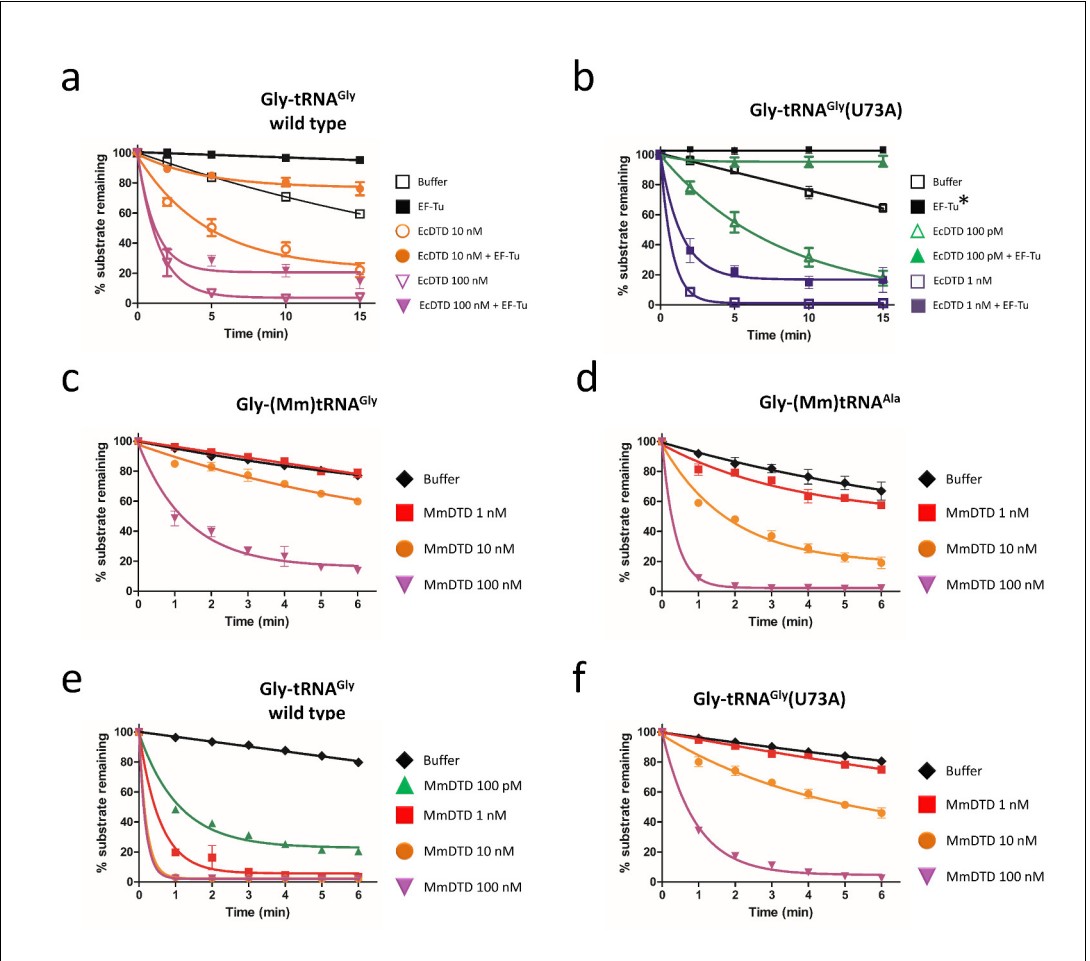

**Figure 3.** Discriminator base predominantly determines the fate of the substrate. Deacylation of (**a**) Gly-tRNA[Gly] and (**b**) Gly-tRNA[Gly](U73A) by EcDTD in the presence or absence of EF-Tu (* indicates the data points are connected through line). Deacylation of (**c**) Gly-(Mm)tRNA[Gly] and (**d**) Gly-(Mm)tRNA[Ala] by various concentrations of MmDTD. Deacylation of (**e**) Gly-tRNA[Gly] and (**f**) Gly-tRNA[Gly](U73A) by various concentrations of MmDTD. Lines indicate exponential decay fit and error bars represent one standard deviation from the mean of at least three independent readings.

DOI: https://doi.org/10.7554/eLife.38232.008

The following source data and figure supplement are available for figure 3:

**Source data 1.** Biochemical data for EF-Tu protection assays and MmDTD deacylations.
DOI: https://doi.org/10.7554/eLife.38232.010

**Figure supplement 1.** Northern blotting showing PfDTD overexpression using IPTG leads to depletion of Gly-tRNA[Gly] while the inactive mutant of PfDTD (A112F) has no effect.
DOI: https://doi.org/10.7554/eLife.38232.009

(*Routh et al., 2016*). While the above data indicated the role of DTD in depleting the cognate Gly-tRNA[Gly] pool, it did not provide direct evidence to that effect. We therefore performed northern blotting analysis by overexpressing the wild-type and a catalytically inactive mutant of DTD (A112F) (*Ahmad et al., 2013*), and monitoring the aminoacylated and free tRNA[Gly] levels. In case of mutant DTD overexpression, the aminoacylated fraction is about 80% of the total tRNA[Gly] pool which is similar to empty vector control. By contrast, overexpression of wild-type DTD causes complete depletion of Gly-tRNA[Gly]. Interestingly, cognate depletion is observed even in the uninduced sample of wild-type DTD, suggesting leaky expression of the wild-type copy from the recombinant plasmid (*Figure 3—figure supplement 1*). These data are suggestive that the toxicity caused by DTD overexpression is linked to the depletion of cellular Gly-tRNA[Gly] levels.

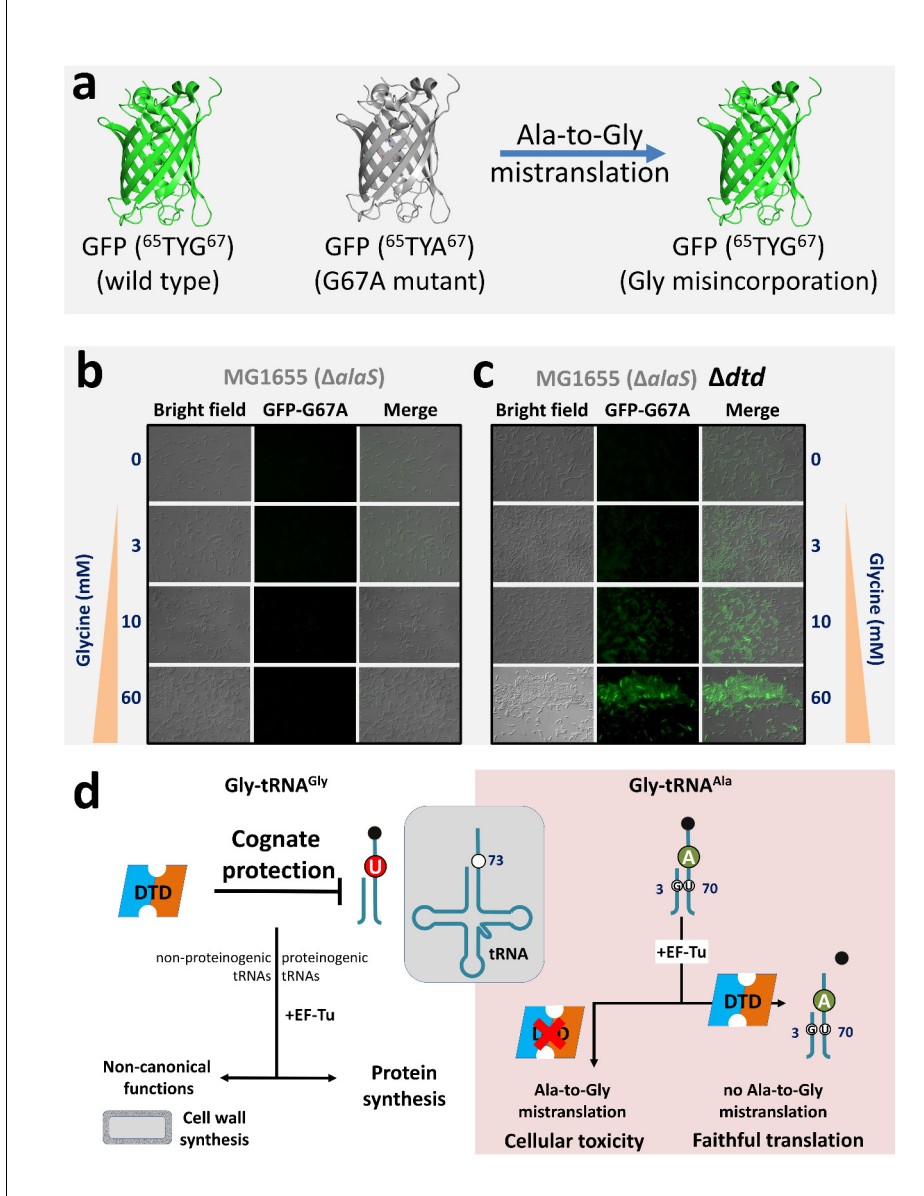

**Figure 4.** DTD avoids glycine misincorporation into proteins. (a) GFP-based fluorescence reporter assay for visualizing alanine-to-glycine mistranslation, wherein the mutant GFP G67A ($^{65}$TYA$^{67}$) will fluoresce only when TYA is mistranslated to TYG. Microscopy images showing GFP fluorescence in *E. coli* at different concentrations of glycine supplementation (b) in the presence and (c) in the absence of DTD. The *E. coli* strain used is MG1655 with editing-defective AlaRS gene (i.e. Δ*alaS*) (*Pawar et al., 2017*). (d) Model showing N73 dichotomy in bacterial tRNA$^{Gly}$ and tRNA$^{Ala}$, enabling protection of the cognate Gly-tRNA$^{Gly}$ (both proteinogenic and non-proteinogenic) predominantly by U73, while effecting efficient removal of the non-cognate Gly-tRNA$^{Ala}$ (having A73 and G3•U70) to prevent alanine-to-glycine mistranslation.

DOI: https://doi.org/10.7554/eLife.38232.011

## G3•U70 is a universal determinant for DTD

Intriguingly, our bioinformatic analysis also revealed that the dichotomy of N73 seen in bacterial tRNA$^{Gly}$ and tRNA$^{Ala}$ is lacking in Archaea and Eukarya, that is both tRNA$^{Gly}$ and tRNA$^{Ala}$ in Archaea and Eukarya harbor A73 (*Figure 1c*). Thus, we envisaged that the difference in eukaryotic DTD's activity on eukaryotic Gly-tRNA$^{Gly}$ and Gly-tRNA$^{Ala}$ will be only about 10-fold due to G3•U70 in tRNA$^{Ala}$ (notably, Archaea lacks canonical DTD and hence the problem of discrimination does not arise at all). To investigate this paradigm, we used DTD and tRNAs from *Mus musculus* (MmDTD and

(Mm)tRNA$^{Gly/Ala}$). Biochemical assays revealed that MmDTD acts on Gly-(Mm)tRNA$^{Gly}$ and Gly-(Mm)tRNA$^{Ala}$ at 100 nM and 10 nM, respectively (*Figure 3c,d*). These results clearly show that G3•U70 recognition by DTD is conserved even in Eukarya, contributing about 10-fold discrimination between eukaryotic tRNA$^{Gly}$ and tRNA$^{Ala}$. They also demonstrate that G3•U70 is the only available tRNA element that enables distinction between eukaryotic tRNA$^{Gly}$ and tRNA$^{Ala}$ by eukaryotic DTD. These findings further suggest that other mechanisms are possibly operating in eukaryotes to enhance the discrimination between Gly-tRNA$^{Gly}$ and Gly-tRNA$^{Ala}$, as discussed later.

## Eukaryotic DTD has inverted N73 specificity

The lack of N73 difference in eukaryotic tRNA$^{Gly}$ and tRNA$^{Ala}$ further prompted us to investigate whether eukaryotic DTD has lost the specificity for N73. To probe the N73 effect on eukaryotic DTD, we performed biochemical assays using MmDTD and Gly-(Ec)tRNA$^{Gly}$. MmDTD acts on U73-containing wild-type Gly-(Ec)tRNA$^{Gly}$ at a concentration of 0.1 nM while it acts on Gly-(Ec)tRNA$^{Gly}$(U73A) mutant with comparable efficiency at a concentration of 10 nM, thus displaying a 100-fold reduction in activity compared to the wild-type substrate (*Figure 3e,f*). The data indicate that N73 does affect eukaryotic DTD but the specificity/preference has switched from purine to pyrimidine. Therefore, eukaryotic DTD is equally modulated by N73 like its bacterial counterpart, but has inverted N73 specificity, that is bacterial DTD prefers purine while eukaryotic DTD prefers pyrimidine as N73; this provides a rationale as to why a mild overexpression of eukaryotic DTD from *Plasmodium falciparum* (PfDTD; eukaryotic DTD) creates more cellular toxicity when compared to that of EcDTD (*Routh et al., 2016*). These findings clearly establish that eukaryotic DTD has also co-evolved with N73 like bacterial DTD, thereby helping to relieve the selection pressure of keeping U/C73 in eukaryotic tRNA$^{Gly}$, as further explained later. Nevertheless, since none of the U/C73-containing eukaryotic tRNAs (*Figure 1c*) is known to be mischarged with glycine or corresponding D-amino acids, the physiological consequence of this switch in N73 specificity remains to be elucidated.

## DTD prevents glycine misincorporation in proteins *in vivo*

One of the major questions that came out of our recent work (*Pawar et al., 2017*) was whether DTD is just a 'recycler' of mischarged tRNAs, as has been shown in the case of D-aminoacyl-tRNAs (*Soutourina et al., 2004*). D-aminoacyl-tRNAs are discriminated against by other cellular 'chiral checkpoints', namely EF-Tu and ribosome (*Bhuta et al., 1981*; *Englander et al., 2015*; *Pingoud and Urbanke, 1980*; *Yamane et al., 1981*), which together with DTD preclude D-amino acid infiltration into the translational machinery. Since Gly-tRNA$^{Ala}$ is not expected to be adequately distinguished from L-Ala-tRNA$^{Ala}$ by EF-Tu and ribosome (*Dale et al., 2004*), we hypothesized that glycine mischarged on tRNA$^{Ala}$ can be misincorporated into the growing polypeptide chain in the absence of DTD, and that DTD's efficient activity on Gly-tRNA$^{Ala}$ due to the presence of A73 and G3•U70 prevents this deleterious outcome. To test this hypothesis *in vivo*, we developed a GFP-based reporter assay (*Figure 4a*). The ability of GFP to fluoresce depends on a key motif ($^{65}$S/TYG$^{67}$) in which mutation of the invariant glycine to any other residue abrogates chromophore formation, resulting in complete loss of fluorescence (*Tsien, 1998*; *Zimmer, 2002*). We expressed the non-fluorescing G67A mutant of GFP in an *E. coli* strain with/without DTD in the editing-defective AlaRS background (*Pawar et al., 2017*). While no GFP fluorescence is observed in cells with the genomic copy of *dtd* gene (*Figure 4b*), the *dtd* knockout strain shows GFP fluorescence upon glycine supplementation, clearly indicating misincorporation of glycine in place of alanine (*Figure 4c*). Moreover, the increase in fluorescence with increasing glycine supplementation in the *dtd*-null *E. coli* strain substantiates the increased toxicity upon glycine supplementation observed in our recent study (*Pawar et al., 2017*). These data clearly reveal that DTD is not merely a 'recycler' of Gly-tRNA$^{Ala}$, but prevents alanine-to-glycine misincorporation in proteins and consequent cellular toxicity because A73 and G3•U70 in the substrate enable efficient editing by the enzyme.

## Discussion

The present work has identified N73 as the major factor that modulates DTD's activity. The invariant U73, which acts as an anti-determinant, is predominantly responsible for ensuring the escape of Gly-tRNA$^{Gly}$ from DTD's unwarranted activity. Strikingly, this protection is essential and significantly higher than that offered by EF-Tu. Notably, EF-Tu's role in conferring protection on Gly-tRNA$^{Gly}$

holds significance only when the substrate contains U73. Such a strong influence of U73 assumes even more importance for non-proteinogenic Gly-tRNA$^{Gly}$, which are required for non-canonical functions (e.g., cell wall synthesis in Gram-positive bacteria) but are not known to interact with EF-Tu or any other protein, except the enzymes (like FemXAB) that utilize it (*Giannouli et al., 2009*; *Katz et al., 2016*). For this achiral species, U73 is the only means of preventing misediting by DTD and consequent cognate depletion. Interestingly, we have now shown that just two tRNA elements—N73 and G3•U70—are sufficient to completely switch DTD's tRNA specificity, thus accounting for the enzyme's 1000-fold difference in activity on Gly-tRNA$^{Gly}$ and Gly-tRNA$^{Ala}$. G3•U70 wobble base pair in the acceptor stem of tRNA$^{Ala}$ is a universal determinant for AlaRS specificity (*Beebe et al., 2008*; *Hou and Schimmel, 1988*; *McClain and Foss, 1988*). This specificity is so strict that G•U at the third base pair position, or in certain cases even at the fourth base pair level, can result in AlaRS charging non-cognate tRNAs with L-alanine (*Kuncha et al., 2018*; *Sun et al., 2016*). We have further shown that G3•U70-based discrimination between tRNA$^{Gly}$ and tRNA$^{Ala}$ by DTD is also conserved in Eukarya. These findings unequivocally demonstrate the obligatory requirement for DTD to use tRNA elements in a major way to discriminate between two cellular substrates, one essential for protein synthesis and the other a misaminoacylated species, carrying the same amino acid, that is achiral glycine. Moreover, the study has also suggested that the high efficiency of bacterial DTD in clearing Gly-tRNA$^{Ala}$ is the plausible reason why it shows significant activity on Gly-tRNA$^{Gly}$, albeit at a 1000-fold less efficiency. This points towards an interesting trade-off between speed and accuracy that exists in the case of DTD's proofreading activity, thereby making DTD uniquely distinct from all other known editing enzymes. It further conforms to the notion that DTD levels in the cell must be tightly regulated.

Surprisingly, our *in silico* analysis has revealed that eukaryotic tRNA$^{Gly}$ and tRNA$^{Ala}$ both contain A73. The lack of N73 difference indicates that G3•U70 is the only discriminatory factor for eukaryotic DTD. It also suggests that in eukaryotes other mechanisms such as spatiotemporal regulation of expression and cellular compartmentalization of DTD may play role in preventing cognate (Gly-tRNA$^{Gly}$) depletion. It is worth mentioning in this context that human DTD has been shown to be enriched in the nuclear envelope region (*Zheng et al., 2009*). Intriguingly enough, our data have revealed that eukaryotic DTD shows a switch in N73 specificity, that is unlike bacterial DTD, the eukaryotic enzyme prefers pyrimidine. However, the physiological relevance of this specificity switch in eukaryotic DTD remains to be elucidated. Moreover, whether such a tRNA-based discriminatory code exists for other standalone proofreading modules such as AlaXs, YbaK and ProXp-ala remains to be probed. Interestingly, such an acceptor stem based tRNA determinants has been shown to exist for other standalone proofreading modules also such as AlaXs, YbaK and ProXp-ala (*Bacusmo et al., 2017*; *Beebe et al., 2008*; *Das et al., 2014*; *Vargas-Rodriguez and Musier-Forsyth, 2013*). These findings also suggest that bacterial DTD and A/G73-containing tRNA$^{Gly}$ or eukaryotic DTD and U/C73-containing tRNA$^{Gly}$ are likely to be incompatible. This can be seen from the delicate balance of the cellular concentrations of DTD and the deleterious effect of its overexpression in *E. coli* (*Routh et al., 2016*).

The uneven, yet interesting, distribution of tRNAs into different discriminator classes based on N73 was used to propose an elegant 'Discriminator Hypothesis' by Donald Crothers about 50 years ago (*Crothers et al., 1972*). This classification also reflects the partitioning of tRNAs based on the chemical nature of the amino acid to be attached, that is A73 class includes tRNAs which code for hydrophobic amino acids, while G73-containing tRNAs code for hydrophilic amino acids (*Crothers et al., 1972*). As can be seen from our current bioinformatic analysis on the discriminator code usage from all three branches of life (*Figure 1c*), N73 is predominantly a purine. It has been already noted that factors like RNase P and CCA-adding enzyme have led to purine bias in N73 in all the three domains of life (*Figure 1c*) (*Burkard et al., 1988*; *Connolly et al., 2004*; *Giegé et al., 1998*; *Hamann and Hou, 1995*; *Hou et al., 2001*; *Puglisi et al., 1994*; *Wende et al., 2015*). In the case of bacteria, the only variations to the purine-based discriminator code are tRNA$^{Gly/His/Cys}$. C73 in prokaryotic tRNA$^{His}$ is essential to retain G$_{-1}$ during processing by RNaseP, while in the case of eukaryotic tRNA$^{His}$, G$_{-1}$ is added latter as a post-transcriptional modification (*Burkard et al., 1988*; *Connolly et al., 2004*). In tRNA$^{Cys}$, U73 is essential for aminoacylation by cysteinyl-tRNA synthetase (*Hamann and Hou, 1995*). Recent evidence shows that CCA-adding enzyme prefers purine as N73 because pyrimidine at that position leads to slow addition of CCA as well as a compromise in the fidelity of CCA addition (*Wende et al., 2015*). The above examples in conjunction with the current

day discriminator code usage (*Figure 1c*) clearly demonstrate that a stronger selection pressure is required to deviate from a purine-based discriminator code. Therefore, it was striking that the tRNA coding for an amino acid which is achiral, having no side chain and of primordial origin is the only bacterial tRNA containing U73 which remained a major anomaly to the discriminator code. The current study has elucidated this hitherto unexplained case of bacterial tRNA$^{Gly}$, wherein the evolutionary selection pressure exerted by DTD for retention of U73 appears stronger than that exerted by any of the other aforesaid factors. It also suggests that eukaryotes have evolved mechanisms that helped in relieving the selection pressure of keeping U73 in tRNA$^{Gly}$, and eukaryotic DTD has co-evolved to switch its N73 specificity accordingly.

Taken together, the present work has added a new dimension to DTD's substrate selectivity, highlighting the role of two key tRNA elements, namely N73 and the G3•U70 wobble base pair. In this respect, it would be interesting to probe the mechanistic underpinnings for DTD's recognition mode using both bacterial and eukaryotic systems to help us understand the underlying cause of switch in N73 specificity. Furthermore, the current study has underscored the physiological importance of N73 dichotomy in tRNA$^{Gly}$ and tRNA$^{Ala}$ that maintains 'glycine fidelity' during translation of the genetic code in Bacteria by preventing misediting of Gly-tRNA$^{Gly}$ as well as misincorporation of glycine in proteins (*Figure 4d*). In the latter context, therefore, DTD is not just a recycler of Gly-tRNA$^{Ala}$, unlike its role in recycling D-aminoacyl-tRNAs (*Soutourina et al., 2004*). The work has brought to the fore the evolutionary selection pressure that the chiral proofreading enzyme has exerted on bacterial tRNA$^{Gly}$ to retain U73. In the primordial scenario, U73 could have been the only factor that ensured utilization of glycine for protein synthesis by precluding the unwarranted activity of DTD-like factor. Our findings have also revealed the co-evolution of eukaryotic DTD via switching of N73 specificity as this selection pressure was released through unknown mechanisms during the 1.5 billion years of evolutionary transition from bacteria to eukaryotes. The study has therefore brought to limelight the deviations that occur in the usage of discriminator base—rare choices of pyrimidine instead of purine—in tRNAs across all life forms, and the need to probe the physiological necessity to deviate from it (*Figures 1c* and *4d*). Considering the possibility that the early tRNAs operated using an acceptor stem–based 'second genetic code' or with mini-helices (*Buechter and Schimmel, 1993*; *Schimmel et al., 1993*), the choice of discriminator base must have played a crucial role during the early evolution of the translational apparatus.

## Materials and methods

### Cloning, expression and purification

The genes encoding *E. coli* DTD and *Thermus thermophilus* EF-Tu, and cDNA encoding *M. musculus* (residues 1–147) were cloned (with C-terminal 6X His-tag), expressed and purified as mentioned in *Ahmad et al. (2013)* and *Kuncha et al. (2018)*. Briefly the proteins were overexpressed in *E. coli* BL21(DE3) by growing the culture to 0.6 OD$_{600}$ at 37°C, followed by induction using 0.5 mM IPTG for 12 hr at 18°C. The cells were harvested by centrifugation at 8000 rpm for 10 min. The harvested cells were lysed and the supernatant was subjected to two-step purification involving affinity-based chromatography followed by size exclusion chromatography (SEC). Affinity purification was done using Ni-NTA column in a solution containing 250 mM NaCl and 100 mM Tris pH 8.0. Imidazole gradient from 10 mM to 500 mM was used to elute the bound protein. Affinity-purified protein was further subjected to SEC to achieve improved purity and homogeneity in a buffer containing 200 mM NaCl and 100 mM Tris pH 7.5. These purified proteins were quantified and stored at −30°C in a buffer containing 50% glycerol. Same procedure was followed for all the proteins except MmDTD which was expressed in *E. coli* BL21-CodonPlus(DE3)-RIL strain.

### Biochemical assays

tRNAs were generated by *in vitro* transcription of genes encoding *E. coli* tRNA$^{Gly/Ala}$ and *M. musculus* tRNA$^{Gly/Ala}$ using MEGAshortscript T7 Transcription Kit (Thermo Fisher Scientific, USA). Various mutants of tRNA$^{Gly}$ were generated using QuickChange XL Site-Directed Mutagenesis Kit (Agilent Technologies, USA). The *in vitro* transcribed tRNAs were 3′ end labelled using CCA-adding enzyme (*Ledoux and Uhlenbeck, 2008*). Glycylation of tRNA$^{Gly}$ and mutants of tRNA$^{Gly}$ (U73A, U73G, U73C, U73A/C70U) were done using *T. thermophilus* glycyl-tRNA synthetase, while tRNA$^{Ala}$ was

glycylated using *E. coli* AlaRS as explained in *Pawar et al. (2017)*. Deacylation assays and EF-Tu protection experiments were performed according to the protocol explained in *Pawar et al. (2017)*. Briefly, deacylation was carried out, in 20 mM Tris pH 7.2, 5 mM dithiothreitol (DTT), 20 mM MgCl$_2$, 0.2 μM of substrate (aminoacylated tRNA), at 30°C with variable concentrations of DTD. EF-Tu activation is done in buffer containing 50 mM HEPES pH 7.2, 20 mM MgCl$_2$, 250 mM NH$_4$Cl, 5 mM DTT, 3 mM phosphoenol pyruvate and pyruvate kinase as explained in *Routh et al., 2016*. EF-Tu protection assays were performed in a buffer containing 2 μM *T. thermophilus* EF-Tu's (of which usually 10–15% is activated (*Cvetesic et al., 2013*; *Sanderson and Uhlenbeck, 2007a*), hence EF-Tu effective concentration is 200–300 nM), 100 mM HEPES pH 7.2, 2.5 mM DTT and variable concentrations of DTD (determined by Bradford assay). A range of enzyme (DTD) concentrations were tested, and, in general, the concentration that gave a gradual deacylation curve was selected for comparison and reporting. GraphPad Prism software was used for curve fitting and every data point represents mean of at least three independent readings. Error bars indicate one standard deviation from the mean.

## Northern blotting

Northern blotting was performed according to the protocol given in *Varshney et al., 1991*. Briefly, 1% of overnight-grown 2 mL primary culture (*E. coli* MG1655Δ*dtd*::Kan cells overexpressing PfDTD wild-type or A112F mutant) was used to initiate 10 mL secondary culture and grown at 37°C till OD$_{600}$ reached 0.6, following which the culture was induced with 0.1 mM IPTG and allowed to grow at 37°C for additional 5–6 hr. The culture was harvested and total RNA was extracted using the general acidic phenol–based method; all the steps were carried out on ice or at 4°C. This was followed by running the total RNA (0.15–0.25 $A_{260}$ unit) on 6.5% acid–urea PAGE at 4°C for 20–24 hr. The RNA from the gel was electroblotted on Hybond$^+$ membrane at 15 V, 3 A for 25 min, and tRNA-$^{Gly}_{GCC}$ was hybridized with [$^{32}$P]-labeled probe (probe sequence: 5'-AGCGGGAAACGAGAC TCGAACTCGC-3'). The probe was labeled using [γ-$^{32}$P]-ATP in a general polynucleotide kinase reaction. The signal from the probe was recorded overnight on image plate and quantified using phosphoimager.

## Strain construction

Wild-type GFP (pBAD18-sfGFP) was a kind gift from Dr. Manjula Reddy's laboratory. Non-fluorescing mutant of GFP was generated by introducing a glycine-to-alanine point mutation at the 67th position (sfGFP-G67A). Editing-defective *alaS E. coli* MG1655 strain (i.e. Δ*alaS*) was created by knocking out the genomic copy of *alaS* and complementing it with triple-mutant (T567F/S587W/C666F) AlaRS gene cloned in pBAD33 vector. The same strategy of making AlaRS editing-deficient was also used in Δ*dtd E. coli* MG1655 strain, thereby creating *dtd*-null strain in AlaRS editing-defective background (i.e. Δ*dtd*Δ*alaS*) (*Pawar et al., 2017*). sfGFP and sfGFP-G67A, cloned in pBAD18 vector, were under the control of an arabinose-inducible promoter.

## Microscopy

To monitor GFP fluorescence in Δ*alaS* (Δ*alaS*/p$_{ara}$::*alaS*-T567F, S587W, C666F) and Δ*dtd*Δ*alaS* (Δ*dtd*, Δ*alaS*/p$_{ara}$::*alaS*-T567F, S587W, C666F) strains were co-expressed with mutant GFP (pBAD18-GFP G67A). Primary cultures were grown at 37°C in LB medium containing 0.002% L-arabinose, 100 mg ml$^{-1}$ ampicillin and 20 μg ml$^{-1}$ chloramphenicol. 3% inoculum was used to initiate 5 mL secondary culture in 1X minimal salts with 0.2% maltose as carbon source and 0.002% L-arabinose; the secondary culture was supplemented with 0, 3, 10 or 60 mM glycine. Cells were immobilized on a thin agarose (1.5%) slide and visualized under a Zeiss Axioimager microscope in DIC (Nomarski optics) and EGFP (Fluorescence) mode. All the experiments were done in biological triplicates.

## Bioinformatic analysis

The prokaryotic tRNA sequences were analyzed using the tRNADB-CE (*Abe et al., 2014*). The numbers were derived using advance pattern search option, wherein anticodon sequence and the $^{73}$NCCA$^{76}$ were used as the search parameters and the rest were set to default. While eukaryotic tRNAs were retrieved from GtRNAdb (http://gtrnadb.ucsc.edu/), only those sequences whose tRNA-

scan score is above 50 were considered for analysis. The consensus sequence logo was obtained by using WebLogo server (http://weblogo.berkeley.edu/logo.cgi).

## Acknowledgements

The authors thank Prof. Umesh Varshney and Ashwin G, Indian Institute of Science, Bengaluru, India for their help with northern blotting. SKK thanks DST-INSPIRE, India, and KS thanks DBT-RA Programme, India, for research fellowships. RS acknowledges funding from 12th Five Year Plan Project BSC0113 of CSIR, India, JC Bose Fellowship of SERB, India, and Centre of Excellence Project of Department of Biotechnology, India. The funding agencies had no role in study design, analysis, decision to publish or preparation of the manuscript.

## Additional information

### Funding

| Funder | Grant reference number | Author |
| --- | --- | --- |
| Department of Science and Technology, Ministry of Science and Technology | DST-INSPIRE | Santosh Kumar Kuncha |
| Department of Biotechnology , Ministry of Science and Technology | DBT-RA | Katta Suma |
| Department of Biotechnology, Ministry of Science and Technology | Centre of Excellence | Rajan Sankaranarayanan |
| Science and Engineering Research Board | J. C. Bose Fellowship | Rajan Sankaranarayanan |

The funders had no role in study design, data collection and interpretation, or the decision to submit the work for publication.

### Author contributions

Santosh Kumar Kuncha, Conceptualization, Formal analysis, Investigation, Methodology, Writing—original draft, Writing—review and editing; Katta Suma, Komal Ishwar Pawar, Satya Brata Routh, Shobha P Kruparani, Formal analysis, Investigation, Methodology, Writing—review and editing; Jotin Gogoi, Formal analysis, Investigation, Methodology; Sambhavi Pottabathini, Investigation, Methodology; Rajan Sankaranarayanan, Conceptualization, Supervision, Writing—original draft, Project administration, Writing—review and editing

### Author ORCIDs

Santosh Kumar Kuncha http://orcid.org/0000-0002-2538-8342
Katta Suma http://orcid.org/0000-0003-4667-8768
Komal Ishwar Pawar http://orcid.org/0000-0002-1968-9851
Jotin Gogoi http://orcid.org/0000-0001-6791-6580
Shobha P Kruparani http://orcid.org/0000-0002-8955-1647
Rajan Sankaranarayanan http://orcid.org/0000-0003-4524-9953

### Decision letter and Author response

Decision letter https://doi.org/10.7554/eLife.38232.015
Author response https://doi.org/10.7554/eLife.38232.016

## Additional files

### Supplementary files

• Transparent reporting form

DOI: https://doi.org/10.7554/eLife.38232.012

## Data availability

Biochemical data is available as a source data file. All other data are included in the manuscript and supporting files.

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
