## [Decision Letter]

Thank you for submitting your article "A discriminator code-based DTD surveillance ensures faithful glycine delivery for protein biosynthesis in bacteria" for consideration by *eLife*. Your article has been reviewed by three peer reviewers, including Jonathan P Staley as the Reviewing Editor and Reviewer #2, and the evaluation has been overseen by Michael Marletta as the Senior Editor. The following individuals involved in review of your submission have agreed to reveal their identity: Michael Ibba (Reviewer #1); Karin Musier-Forsyth (Reviewer #3).

The reviewers have discussed the reviews with one another and the Reviewing Editor has drafted this decision to help you prepare a revised submission.

Summary:

This "Research advance" follows on a recent *eLife* publication from the same group showing that DTD proofreads, in addition to its broad role in deacylating D-aminoacylated tRNAs, by deacylating Gly-tRNA^Ala^. In this manuscript, the authors investigate the mechanistic basis for the 1000-fold discrimination by DTD between mischarged Gly-tRNA^Ala^ and Gly-tRNA^Gly^ in *E. coli*. Here, the authors show intriguingly that 100-fold of the specificity can be accounted for solely by the identity of the so-called discriminator base. Whereas this base is generally a purine, with adenine indicating aliphatic residues and guanine indicating polar residues, in tRNA^Gly^ this base is an exception, as a uracil. In a key experiment, the authors show that mutation of this uracil to adenine or guanine in tRNA^Gly^ renders Gly-tRNA^Gly^ sensitive to (inappropriate) deacylation. Thus, the authors have both rationalized an unusual exception to the discriminator base rules and identified a significant determinant of DTD specificity. The authors further show that in eukaryotes, where the discriminator base in tRNA^Gly^ has shifted to an adenine, the specificity of DTD has switched to prefer purine over uracil, and as a consequence, DTD only imposes a 10-fold reduction in mischarged Gly-tRNA^Ala^. Lastly, in a clever experiment, the authors demonstrate the physiological significance of DTD proofreading of Gly-tRNA^Ala^ by showing that in the absence of DTD, glycine can be mis-incorporated into a key position of GFP, conferring fluorescent functionality to the protein. In the past, DTD was thought to only act on chiral amino acids and to only recognize the amino acid component of aminoacyl-tRNA. Here, the authors show physiological relevance in achiral proofreading for protein synthesis and define a key and intriguing element in the tRNA body that impacts DTD specificity. This work will be of wide interest to those interested in translation and/or mechanisms of specificity and builds very interestingly on the original Crothers Discriminator Hypothesis.

Essential revisions:

1) In subsection “N73 and G3•U70 have an additive effect on DTD’s activity“, the authors assert that they have established DTD only interacts with the acceptor stem, a conclusion that would require additional biophysical data. What has been shown is that the functionally important interactions are confined to the acceptor stem.

2) The idea of "reciprocal" evolution is confusing and ultimately unnecessary. The study clearly shows things work differently in eukaryotes, although how is still unclear, and it would best be left at that.

3) Many experimental details are missing in the Materials and methods section. The concentration of aminoacyl-tRNA substrates is not reported anywhere in the paper. The deacylations performed in the presence of EF-Tu do not report a concentration of EF-Tu. The use of different DTD concentrations is reported, but how were these determined? What were the conditions of the deacylation reactions?

4) The results are qualitative and it is not clear that the quantitative results reported in Supplementary Table 1 are valid. The Km is not reported and the enzyme concentration should be significantly below the Km for this analysis to be valid. In addition, the equation used to calculate *k*_obs_ should only be applied to fit a reaction performed under single turnover conditions for valid comparisons. Was the E in significant excess over tRNA? Was the buffer background rate subtracted prior to calculating the rates?

5) The authors state: "… cognate depletion is observed even in the uninduced sample of wild-type DTD, suggesting leaky expression of the wild-type copy from the recombinant plasmid. These data clearly show that the toxicity caused by DTD overexpression is due to the depletion of cellular Gly-tRNA^Gly^ levels, thus warranting a strict control on endogenous levels of DTD." The Northern blot showed complete depletion of Gly-tRNA^Gly^ even without induction. So, shouldn't this leaky expression also cause cell toxicity?

[Editors' note: further revisions were requested prior to acceptance, as described below.]

Thank you for resubmitting your work entitled "A discriminator code-based DTD surveillance ensures faithful glycine delivery for protein biosynthesis in bacteria" for further consideration at *eLife*. Your revised article has been favorably evaluated by Michael Marletta (Senior Editor) and a Reviewing Editor.

The manuscript has been improved but there are some remaining issues that need to be addressed before acceptance, as outlined below:

1) Regarding essential revision #4, because the reactions are not performed under single turnover conditions, the equation for *k*_obs_ is not valid. Consequently, the authors need to remove Supplementary Table 1.

2) Regarding essential revision #5, the authors have not provided a satisfactory answer. That the authors "believe" that with leaky expression the Gly-tRNA^Gly^ produced is just sufficient for translation is not an acceptable response to the concern; while this is a reasonable explanation for the discrepancy, the data do not support this explanation, presumably due to the insensitivity of the assay to low levels of Gly-tRNA^Gly^. The authors have two options. First, they can repeat the assay and prove that in fact with leaking expression there are low levels of Gly-tRNA^Gly^. Second, omit the premature conclusion that DTD over expression is due to depletion of this tRNA.

---

## [Author Response]

Essential revisions:1) In subsection “N73 and G3•U70 have an additive effect on DTD’s activity“, the authors assert that they have established DTD only interacts with the acceptor stem, a conclusion that would require additional biophysical data. What has been shown is that the functionally important interactions are confined to the acceptor stem.

As correctly pointed out by the reviewer(s), we have not provided any direct evidence that DTD interacts with only the acceptor stem of tRNA. However, unlike other proteins (aaRS and EF-Tu) which interact with tRNA, DTD is smaller in size (16-18 kDa) and a simple modeling suggests that the recognition is not expected to go beyond the tRNA acceptor stem. Nevertheless, as we have no direct evidence, now the statement has been appropriately modified in the revised version of the manuscript, as suggested by the reviewer(s) and given below:

“These results also clearly establish that functionally important interactions of DTD are confined only to the acceptor stem of tRNA.”

2) The idea of "reciprocal" evolution is confusing and ultimately unnecessary. The study clearly shows things work differently in eukaryotes, although how is still unclear, and it would best be left at that.

By “reciprocal” evolution of N73 and DTD in Bacteria and Eukarya, we just wanted to underscore the fact that bacterial DTD acts more efficiently on purine at N73 when compared to pyrimidine at that position, eukaryotic DTD shows exactly the reverse phenomenon. However, since this is confusing and possibly unnecessary, we have removed it from the revised version of the manuscript, as suggested by the reviewer(s).

3) Many experimental details are missing in the Materials and methods section. The concentration of aminoacyl-tRNA substrates is not reported anywhere in the paper. The deacylations performed in the presence of EF-Tu do not report a concentration of EF-Tu. The use of different DTD concentrations is reported, but how were these determined? What were the conditions of the deacylation reactions?

The instruction to authors as mentioned for ‘Research Advance’ suggests not to add a detailed Materials and methods and hence we left out some of the details mentioned in our previous work by Pawar et al., *eLife*, 2017. However, since the reviewer(s) deem it appropriate to incorporate some of the crucial details (such as substrate concentration) in the current manuscript as well, we have now mentioned them in the “Materials and methods” section of the revised manuscript. The enzyme concentration was determined by Bradford assay (further validated by measuring OD_280_ using NanoDrop 1,000 Spectrophotometer (Thermo Scientific)), a range of enzyme concentrations were tested, and in general, the concentration that gave a gradual deacylation curve was selected for comparison and reporting. This too has now been mentioned in the revised manuscript.

4) The results are qualitative and it is not clear that the quantitative results reported in Supplementary Table 1 are valid. The Km is not reported and the enzyme concentration should be significantly below the Km for this analysis to be valid. In addition, the equation used to calculate k_obs_ should only be applied to fit a reaction performed under single turnover conditions for valid comparisons. Was the E in significant excess over tRNA? Was the buffer background rate subtracted prior to calculating the rates?

We agree with the reviewer(s) that the results are qualitative. And therefore, the values given are purely indicative. Since higher concentration of DTD leads to rapid deacylation of substrate we had to use lower concentration of enzyme to get gradual deacylation curves. Moreover, the *k*_obs_ in our equation simply signifies the first-order decay constant, as the curves in the graphs themselves suggest that deacylations at the given enzyme concentrations follow first-order exponential decay. That is the reason for us to keep this table in the supplementary information. The buffer deacylation rate (now mentioned in supplementary table 1) was subtracted to obtain the effective rate of enzymatic deacylation, and this has now been explicitly mentioned in the manuscript. However, if the reviewer(s) still feels that the results need not be quantified in this manner we can remove the Supplementary Table 1.

5) The authors state: "… cognate depletion is observed even in the uninduced sample of wild-type DTD, suggesting leaky expression of the wild-type copy from the recombinant plasmid. These data clearly show that the toxicity caused by DTD overexpression is due to the depletion of cellular Gly-tRNA^Gly^ levels, thus warranting a strict control on endogenous levels of DTD." The Northern blot showed complete depletion of Gly-tRNA^Gly^ even without induction. So, shouldn't this leaky expression also cause cell toxicity?

Our northern blot data do suggest a leaky expression of DTD as Gly-tRNA^Gly^ levels are depleted even in the uninduced condition. However, in our previous study (Routh et al., 2016), we could see toxicity only upon induction. We believe that the level of DTD due to leaky expression does not allow any accumulation of Gly-tRNA^Gly^ in the cell (hence, Gly-tRNA^Gly^ band is not visible in the blot), but the Gly-tRNA^Gly^ produced is just sufficient for translation to continue and prevent toxicity. However, upon induction, this delicate balance is disturbed causing toxicity due to enhanced depletion of Gly-tRNA^Gly^. This data is further strengthened by the fact that in the case of DTD inactive mutant A112F (Adenine binding pocket mutant) the Gly-tRNA^Gly^ levels are unperturbed.

[Editors' note: further revisions were requested prior to acceptance, as described below.]

The manuscript has been improved but there are some remaining issues that need to be addressed before acceptance, as outlined below:1) Regarding essential revision #4, because the reactions are not performed under single turnover conditions, the equation for k_obs_ is not valid. Consequently, the authors need to remove Supplementary Table 1.

As suggested by you, Supplementary Table 1 has now been removed from the revised manuscript as well as the relevant information from the Materials and methods.

2) Regarding essential revision #5, the authors have not provided a satisfactory answer. That the authors "believe" that with leaky expression the Gly-tRNA^Gly^ produced is just sufficient for translation is not an acceptable response to the concern; while this is a reasonable explanation for the discrepancy, the data do not support this explanation, presumably due to the insensitivity of the assay to low levels of Gly-tRNA^Gly^. The authors have two options. First, they can repeat the assay and prove that in fact with leaking expression there are low levels of Gly-tRNA^Gly^. Second, omit the premature conclusion that DTD over expression is due to depletion of this tRNA.

We agree that the discrepancy between the uninduced and induced WT DTD in the northern blotting is probably due to the sensitivity of the experiment. Small amounts Gly-tRNA^Gly^ remaining in the uninduced condition but undetectable in our assay seems to be sufficient for translation to go on without causing cellular toxicity. Hence, the conclusion has now been modified as following:

“These data are suggestive that the toxicity caused by DTD overexpression is linked to the depletion of cellular Gly-tRNA^Gly^ levels”.